# Distinct miRNA Signatures and Networks Discern Fetal from Adult Erythroid Differentiation and Primary from Immortalized Erythroid Cells

**DOI:** 10.3390/ijms22073626

**Published:** 2021-03-31

**Authors:** Panayiota L. Papasavva, Nikoletta Y. Papaioannou, Petros Patsali, Ryo Kurita, Yukio Nakamura, Maria Sitarou, Soteroulla Christou, Marina Kleanthous, Carsten W. Lederer

**Affiliations:** 1Department of Molecular Genetics Thalassemia, The Cyprus Institute of Neurology and Genetics, Nicosia 2371, Cyprus; panayiotap@cing.ac.cy (P.L.P.); nikolettap@cing.ac.cy (N.Y.P.); petrospa@cing.ac.cy (P.P.); marinakl@cing.ac.cy (M.K.); 2Cyprus School of Molecular Medicine, Nicosia 2371, Cyprus; 3Cell Engineering Division, RIKEN BioResource Center, Tsukuba, Ibaraki 305-0074, Japan; r-kurita@jrc.or.jp (R.K.); yukio.nakamura@riken.jp (Y.N.); 4Thalassemia Clinic Larnaca, Larnaca General Hospital, Larnaca 6301, Cyprus; msitarou@yahoo.gr; 5Thalassemia Clinic Nicosia, Archbishop Makarios III Hospital, Nicosia 1474, Cyprus; snchrthalcl@cytanet.com.cy

**Keywords:** microRNA, non-coding DNA, small RNA sequencing, erythropoiesis, CD34+, hematopoietic stem cell, gene regulatory network, long non-coding RNA, competing endogenous RNA, enrichment analysis, developmental regulation

## Abstract

MicroRNAs (miRNAs) are small non-coding RNAs crucial for post-transcriptional and translational regulation of cellular and developmental pathways. The study of miRNAs in erythropoiesis elucidates underlying regulatory mechanisms and facilitates related diagnostic and therapy development. Here, we used DNA Nanoball (DNB) small RNA sequencing to comprehensively characterize miRNAs in human erythroid cell cultures. Based on primary human peripheral-blood-derived CD34+ (hCD34+) cells and two influential erythroid cell lines with adult and fetal hemoglobin expression patterns, HUDEP-2 and HUDEP-1, respectively, our study links differential miRNA expression to erythroid differentiation, cell type, and hemoglobin expression profile. Sequencing results validated by reverse-transcription quantitative PCR (RT-qPCR) of selected miRNAs indicate shared differentiation signatures in primary and immortalized cells, characterized by reduced overall miRNA expression and reciprocal expression increases for individual lineage-specific miRNAs in late-stage erythropoiesis. Despite the high similarity of same-stage hCD34+ and HUDEP-2 cells, differential expression of several miRNAs highlighted informative discrepancies between both cell types. Moreover, a comparison between HUDEP-2 and HUDEP-1 cells displayed changes in miRNAs, transcription factors (TFs), target genes, and pathways associated with globin switching. In resulting TF-miRNA co-regulatory networks, major therapeutically relevant regulators of globin expression were targeted by many co-expressed miRNAs, outlining intricate combinatorial miRNA regulation of globin expression in erythroid cells.

## 1. Introduction

Hematopoiesis is orchestrated by a complex and multidimensional regulatory network that controls blood cell formation in the bone marrow (BM) [1]. The identification of a pool of multipotent hematopoietic stem and progenitor cells (HSPCs) in the BM, typically based on the use of the CD34 as cell surface marker [2,3], allowed the subsequent discovery of distinct, phenotypically defined compartments in the hematopoietic differentiation hierarchy, marked by characteristic changes in transcriptional profiles and key transcription factors (TFs) [4]. The application of high-throughput technologies, like genomic, transcriptomic, proteomic, and other “omic” technologies, allowed identification of lineage-specific expression profiles and study of cellular dynamics and regulatory mechanisms controlling human hematopoietic homeostasis and cell specification [5,6]. The proliferation and differentiation of HSCs down the erythroid lineage, called erythropoiesis, is thus one of the best-characterized developmental models, yet many of its aspects remain obscure [7].

MicroRNAs (miRNAs) are small non-coding RNAs (sncRNAs) of 20–23 nucleotides that posttranscriptionally modulate gene expression, mainly via degradation of mRNAs of target genes and/or inhibition of translation [8,9], although they can also upregulate translation or control transcription [10,11,12]. They represent an axis of regulation of erythropoiesis, fine-tuning the expression of key TFs that control cell fate, with mounting evidence for their additional role in other, more complex competing endogenous RNA (ceRNA) networks [13,14]. An ongoing effort to complement mostly microarray-based high-throughput studies and computational predictions with functional studies of individual miRNAs draws an image of complex modular action for miRNAs, characterized by their involvement in a multitude of highly intertwined processes and by interactions with proteins and with coding and non-coding transcripts.

Landmark studies on miRNAs during erythropoiesis catalogued differential miRNA expression in in vitro erythroid differentiation of human progenitors [15] and revealed progressive downregulation and targeting of the key erythroid receptor c-KIT by the miR-221/222 cluster [16], progressive downregulation of miR-221, miR-223, and miR-155, upregulation of GATA1-driven high-level expression of miR-144 and miR-451 [17,18,19], and LMO2-targeting and progressive downregulation of miR-223 [20]. Subsequently, several miRNAs were shown to be involved in normal and abnormal erythropoiesis or in its accompanying processes, such as iron metabolism and hemoglobin synthesis [21]. Importantly, β-hemoglobinopathies are clinically prominent and are typically ameliorated by elevated expression of the β-like γ-globin, which may substitute β-globin chains of adult hemoglobin (HbA; α_2_β_2_) to form fetal hemoglobin (HbF; α_2_γ_2_) [22,23]. This observation has fueled decades of exceptionally detailed characterization of the perinatal fetal-to-adult switch from β- to γ-globin, which nevertheless is still not fully understood [24,25]. Therefore, further delineation of the underlying mechanisms and the role of miRNAs and other non-coding RNAs (ncRNAs) in hemoglobin switching will facilitate the design of alternative or supplementary therapies for β-hemoglobinopathies. Although several studies have already implicated specific miRNAs in hemoglobin switching, manipulation of single miRNAs has yet to achieve therapeutically relevant HbF reactivation [26,27,28,29]. Based on our growing understanding of normal and pathological miRNA action in networks of multiple-to-multiple rather than one-to-one interactions, this may instead be achieved by concerted manipulation of multiple functionally related miRNAs [30,31,32].

In the present study, we employed small RNA sequencing to explore sncRNA and in particular miRNA expression in ex vivo human erythroid cell cultures from normal peripheral blood (PB)-derived CD34+ HSPCs (hCD34+), as well as in human umbilical cord blood-derived erythroid progenitor (HUDEP) cell lines (HUDEP-1 and HUDEP-2) during erythroid differentiation. To this end, hCD34+, HUDEP-1, and HUDEP-2 cells were expanded, differentiated, characterized, and analyzed for small RNA expression at early and late stages of erythroid commitment. Whereas hCD34+ cells are relevant as the hematopoietic stem and progenitor cell population used for therapeutic transplantation, the HUDEP cell lines are now used by many hematological research groups and currently represent an invaluable model for the study of erythropoiesis and therapy development [33,34,35,36,37,38]. Since their establishment by Kurita et al. based on lentiviral transduction of the doxycycline-inducible human papillomavirus 16 (HPV16)-E6/E7 expression system, HUDEP cells have overwhelmingly replaced other cell lines, such as K562, MEL, and UT-7/EPO, in the analysis of erythropoiesis, owing to their superior similarity to human primary cells. Notably, HUDEP-2 and HUDEP-1 show selective upregulation of β-globin and γ-globin expression, respectively, and BCL11A, a major repressor of γ-globin, is abundantly expressed only in HUDEP-2 cells [39]. Based on these characteristics, global small RNA sequencing of both HUDEP cell lines for comparison in erythropoiesis with one another and with adult primary erythroid cells will provide more authentic insights than those obtained from other erythroid cell lines [18,40,41] and may prove highly informative for the role and possible therapeutic exploitation of detected miRNAs.

In this first global sncRNA study on HUDEP cells, the parallel analysis of miRNA expression in primary erythroid cells and HUDEP erythroid cell lines identifies miRNA signatures associated with erythropoiesis, reveals differences between primary cells and their immortalized equivalent, and provides pointers to miRNAs, miRNA target genes, such as TFs, and pathways mechanistically linked to globin switching. Furthermore, predicted interactions between identified miRNAs and known, erythroid-specific long non-coding RNAs (lncRNAs) indicate evaluable crosstalk between small and long ncRNAs in our erythroid cell cultures.

## 2. Results

The experimental design for this study, including time points of analyses and culture conditions, is illustrated in Figure 1.

### 2.1. Flow Cytometry-Based Cell-Surface Protein Analysis Confirms Consistent Erythroid Differentiation in Cultures

Erythroid differentiation in cell culture systems was assessed by flow cytometry analysis of cell surface protein expression at three time points: early-stage (_E, day 0 of erythroid differentiation), intermediate stage (_I, days 3 and 4 of erythroid differentiation for hCD34+ and HUDEP cells, respectively) and late-stage (_L, days 6 and 9 of erythroid differentiation for hCD34+ and HUDEP cells, respectively). While excluded from sequencing analysis, the intermediate-stage samples served to monitor and confirm progress and comparability of erythropoiesis. Two sets of proteins, CD44 and CD49d (integrin alpha 4) as early differentiation markers and CD235a (glycophorin A, GpA) and CD71 (transferrin receptor) as late differentiation markers, were evaluated, to track erythropoiesis by their differential expression. The corresponding histograms, comparing the expression of markers with the progression of erythropoiesis, are shown in Figure 2a. The observed dynamic changes of expression revealed an overall normal and comparable progression of erythroid differentiation in hCD34+ and HUDEP cells. As expected, expression levels of early-stage markers (CD44 and CD49d) declined upon induction of differentiation, in line with previous reports of their progressive decrease from proerythroblast to reticulocyte stages [42,43]. In our cultures, CD71 rapidly increased from early to intermediate stages, but then declined towards late stages of erythropoiesis, also in agreement with earlier studies, where CD71 was consistently found to increase 3- to 4-fold from proerythroblast to basophilic and polychromatophilic erythroblast stages and then return to proerythroblast level in late-stage orthochromatophilic erythroblasts [43,44]. Finally, CD235 as the major intrinsic protein of the human erythrocyte membrane [45], steadily increased from early to late stages in all cultures, although with a higher baseline expression in early-stage HUDEP cells. The latter observation concurs with the basophilic erythroblast gene expression profile of undifferentiated HUDEP cells previously documented by others [46].

### 2.2. Morphological Changes Characterize Erythroid Differentiation in Cultures

Morphologically, both primary hCD34+ cells and HUDEP erythroid cell lines progressed through the recognizable erythroblastic stages, with polychromatophilic erythroblasts and fewer orthochromatophilic erythroblasts already appearing by day 3 for hCD34+ and by day 4 for HUDEP cells (_I). At the end of erythroid differentiation (_L), hemoglobin-producing cells comprised the majority of cells in hCD34+, HUDEP-1, and HUDEP-2 cell cultures (96.0%, 98.4%, and 99.8%, respectively). At this stage, orthochromatophilic erythroblasts with pyknotic nuclei and enucleated cells prevailed in primary cell cultures, whereas poly- and orthochromatophilic erythroblasts were the predominant cells in HUDEP cell cultures, in the absence of enucleation (Figure 2b). Differential counting of erythroid subpopulations for _E and _L samples (total numbers of cells counted were 1647 for CD34_E, 1047 for HUDEP1_E, 1995 for HUDEP2_E, 1670 for CD34_L, 908 for HUDEP1_L, and 1179 for HUDEP2_L) indicated a greater differentiation potential of hCD34+ cells as shown by the cell fraction of late-stage erythroid differentiation (orthochromatophilic erythroblasts and reticulocytes); however, differences between cell types were not statistically significant (Figure 2c,d). Moreover, cell death, as evaluated by trypan blue assay, was higher in late-stage HUDEP-1 and HUDEP-2 cultures (≈67% and ≈52%, respectively) compared to late-stage primary cell cultures (15%). The above observations are attributed to the documented inherent inefficiency of differentiation of HUDEP cells and the reduced cell viability accompanying the withdrawal of doxycycline in cultures [36,39].

### 2.3. Reversed-Phase High-Performance Liquid Chromatography (RP-HPLC) Analysis Verifies Adult and Fetal Globin Expression Profiles of Differentiated Cells

Separation of globin chains by RP-HPLC was also performed in late-stage samples for hCD34+, HUDEP-1, and HUDEP-2 cells. As expected, hCD34+ cells and HUDEP-2 cells exhibited an adult globin expression profile with α- and β-globin production, whereas HUDEP-1 had a fetal profile with α- and predominantly γ-globin production (Figure 2e).

### 2.4. Small-RNA Profiling during Erythroid Differentiation Detects Novel and Known miRNAs and Delineates Cell Types and Stages for HUDEP and Primary hCD34+ Cells

To identify small RNAs expressed during erythroid differentiation, we used DNA Nanoball (DNB) sequencing technology to sequence small RNAs from total RNA of hCD34+, HUDEP-1, and HUDEP-2 cells at two distinct time points of erythroid differentiation: early-stage (on day 0 for all cell sources and henceforth labeled _E) and late-stage (on days 6 and 9 for hCD34+ and HUDEP cells, respectively, and henceforth labeled _L). Three biological replicates in each stage were analyzed, labeled A, B, and C.

The proportions of reads assigned to known miRNAs (“mature” and “precursor”) and other kinds of sncRNAs (e.g., piwi-interacting RNAs (piRNAs), small nuclear RNAs (snRNAs), small nucleolar RNAs (snoRNAs), small interfering RNAs (siRNAs), transfer RNAs (tRNAs), ribosomal RNAs (rRNAs)), are summarized as pie charts in (Appendix A); for the number of known and novel miRNAs and piRNAs detected for each sample, see Appendix A. In total, we measured the expression of 1707 known miRNAs assigned to 583 miRNA families and identified 2138 novel miRNAs based on the predicted secondary structure of their potential precursors. Out of these, 538 known miRNAs and 210 novel miRNAs had an average read count number above 20 in tested samples, while more than 85 miRNAs showed more than 1000 reads in each sample. Only mature miRNAs and no miRNA precursors or other classes of sncRNAs were considered for further analyses in the context of this study.

MiR-126-3p was the most highly expressed miRNA in early-stage hCD34+ and HUDEP-2 samples, contributing, respectively, 30.39% and 27.61% of all reads mapping to known and novel miRNAs, whereas miRNA-92a-3p was the most highly expressed miRNA in early-stage HUDEP-1 cells, contributing 16.05% of all reads. MiR-451a was the most highly expressed miRNA in all late-stage samples, contributing 15.85%, 16.59%, and 13.71% of all reads in hCD34+, HUDEP-1, and HUDEP-2 samples, respectively (Appendix A).

Most of the predicted novel miRNAs had relatively low abundances and were identified only in few samples (Appendix A). Thirty-seven novel miRNAs were expressed in all tested samples. Among them, novel_mir1 was the most abundant, its average number of reads contributing 50.45% and 1.37% of all reads for novel and for all (novel and known) miRNAs, respectively. Novel_mir1 was mapped to chr17:30306982_30307057, based on the predicted secondary structure of its potential precursor.

For a global view of miRNA expression in our cell culture systems, we performed principal component (PC) analysis of known and novel expressed miRNAs (Figure 3a). The first PC (PC1), accounting for 28.6% of the variation in the dataset, corresponds to the stage of erythroid differentiation (early vs. late), indicating an overall pattern of temporal changes in expression, whereas the second PC (PC2), accounting for 15.5% of the variation, corresponds to the cell type. Notably, PC2 grouped both adult-type cell sources, HUDEP-2 and hCD34+ samples, together, while separating fetal-type HUDEP-1 samples.

For pairwise comparisons of interest (same-stage expression between cell types, and late- vs. early-stage expression for each cell type) we performed differential expression analysis of all known and novel miRNAs using DEGseq [47]. For this analysis, DE miRNAs were defaulted as miRNAs with a false discovery rate (FDR) < 0.001 and a fold change (FC) ≥ 2 up or down (|log_2_ FC| ≥ 1). Of note, all log_2_ FCs cited in this article were statistically significant with a *p*-value <1.00 × 10^−300^, unless stated otherwise. Hierarchical clustering of all known and novel DE miRNAs confirmed the result of PCA, showing the relatedness between hCD34+ and HUDEP-2 cells as captured at both early and late stages of erythroid differentiation (Figure 3b,c). When samples were analyzed for changes in miRNA expression upon induction of erythroid differentiation (_L vs. _E), HUDEP-1 and HUDEP-2 cells clustered together and separately from hCD34+, although the inter-cluster variance was, in that case, low (Figure 3d), indicating an overall common erythroid differentiation signature for all three cell types. The top 60 most variable known miRNAs across cell types and stages, as defined by the standard deviation (SD) of expression, were sorted by descending average expression and plotted in a heatmap (Appendix A).

### 2.5. Temporal miRNA Expression Profiling Shows Dynamic Regulation of Erythroid Differentiation

To identify miRNAs that are dynamically expressed during erythropoiesis, we looked for known and novel DE miRNAs between _L and _E samples. Evidently, more miRNAs were downregulated than upregulated in the late stages of erythroid differentiation in all cell cultures (Figure 4a). Consistent with progressive nuclear and chromatin condensation and downregulation of global gene expression after erythroid lineage commitment [48,49], our data revealed a restricted pattern of global miRNA expression and a reciprocal increase of the expression of individual lineage-specific miRNAs in terminal stages of erythropoiesis. This finding mirrors at the miRNA level what has been observed at the mRNA level, i.e., that less differentiated cells retain a poised cell state that is characterized by transcriptional complexity [50], and indicates active regulation of the cell state by a large repertoire of miRNAs.

Known miRNAs were tested for differential abundance in pairwise comparisons of late- vs. early-stage expression for hCD34+, HUDEP-1, and HUDEP-2 cells, respectively (Figure 4b–d). Several miRNAs showing significant upregulation upon induction of erythroid differentiation in our study were previously identified as lineage-specific miRNAs in other studies (see also Appendix A and Appendix A). Among them, miR-451a was the most highly expressed miRNA in late samples for all three cell types and significantly upregulated during erythroid differentiation, showing an _L vs. _E log_2_ FC of 4.98, 2.04, and 3.8, in hCD34+, HUDEP-1, and HUDEP-2 cells, respectively. Other erythroid-specific miRNAs, such as miR-15b-5p, miR-16-5p, miR-96-5p, and miR-22-3p, were also significantly upregulated in CD34_L (in line with published data [18,41,51]) and HUDEP2_L cells, whereas only 15b-5p was upregulated, albeit insignificantly, in HUDEP1_L cells. Two miRNAs recently identified as associated with erythropoiesis, miR-182-5p and miR-4732-3p [52], were validated as such in our study, showing an _L vs. _E log_2_ FC of 4.31 and 3.36 (for miR-182-5p) and 3.85 and 3.7 (for miR-4732-3p) in hCD34+ and HUDEP-2 cells, respectively. Among several miRNAs downregulated during differentiation, miR-223-5p significantly decreased from _E to _L in hCD34+ and HUDEP-2 cells (_L vs. _E log_2_ FC of −1.39 and −3.12, respectively), and decreased, albeit insignificantly, in HUDEP-1 cells. Both miRNAs targeting c-KIT, miR-221-3p and miR-222-3p [16], were highly expressed in CD34_E cells and significantly downregulated after erythroid cell commitment, but interestingly, only faintly expressed at both early and late stages of erythroid differentiation of HUDEP-1 and HUDEP-2 cells. Finally, miR-181a-3p, previously shown to decrease at late stages of erythropoiesis, de-repress Exportin 7 and regulate final stages of normal erythroid development, such as chromatin condensation and enucleation in murine erythroid cells [53], was significantly downregulated only in hCD34+ and HUDEP-2 cells (_L vs. _E log_2_ FC = −3.98 and −5.84, respectively). A Venn diagram of late vs. early DE known miRNAs in hCD34+, HUDEP-1, and HUDEP-2 cells revealed a distinct set of 157 common miRNAs, likely representing the miRNAs most relevant to erythroid differentiation in general (Figure 4e and Appendix A).

Novel miRNAs were also tested for differential abundance in the same pairwise comparisons of _L vs. _E for each cell type (Appendix A). The most abundant novel miRNA, novel_mir1, was significantly downregulated upon induction of erythroid differentiation in all three cell culture systems (_L vs. _E log_2_ FC = −1.51, −1.15 and −1.55 in hCD34+, HUDEP-1, and HUDEP-2 cells, respectively). Novel_mir1055, the second most abundant novel miRNA, was significantly upregulated in hCD34+ and HUDEP-2 cells (_L vs. _E log_2_ FC = 1.98 and 1.44, respectively). Another miRNA, novel_mir603, increased sharply with erythroid maturation in all cell cultures, showing an _L vs. _E log_2_ FC of 4.00, 3.63, and 4.56, in hCD34+, HUDEP-1, and HUDEP-2 cells, respectively.

By considering known miRNAs DE between early and late stages of erythroid differentiation, and their target genes, we identified significantly enriched biological pathways in hCD34+, HUDEP-1, and HUDEP-2 cells using Reactome [54] (Appendix A). This confirmed pathways at the miRNA level that had been highlighted in previous conventional transcriptomic studies of HSPC proliferation and erythroid differentiation, including the EGFR, the SCF-KIT, the MAPK, and the PI3K-Akt signaling pathways [55,56,57], and also indicated novel pathways, such as signaling by ErbB2 and by Wnt.

### 2.6. Characterisation of the miRNA Transcriptome Highlights Differences between Primary hCD34+ Cells and HUDEP Cell Lines

The characterization and comparative analysis of the small-RNA transcriptome in hCD34+ and HUDEP cells showed an overall common erythroid differentiation signature, which vindicates the use of HUDEP cells as informative in vitro models for human erythropoiesis, also for the analysis of sncRNAs. Differential expression analysis of known and novel miRNAs in same-stage HUDEP-1 vs. hCD34+ and HUDEP-2 vs. hCD34+ confirmed the greater relatedness between hCD34+ and HUDEP-2 cells but also revealed nuanced differences that must be taken into consideration when those models are used to evaluate novel therapeutic targets and associated molecular mechanisms.

We focused on differential expression analysis of known miRNAs in HUDEP2_L vs. CD34_L cells as the most informative comparison for an assessment of the fidelity of findings from HUDEP-2 cells in functional studies of miRNAs (Figure 5a and Appendix A). Differential expression compared to hCD34+ cells would indicate potentially artefactual observations in HUDEP-2 cells (see also Appendix A and Appendix A). In total, 172 and 242 known miRNAs were significantly upregulated in HUDEP2_L and CD34_L cells, respectively. By mapping those miRNAs to the genomic location of their precursor sequences using Bowtie 2 (Figure 5b), we identified five miRNAs of the miR-106a/363 cluster located on the Xq26.2 chromosome as significantly upregulated in HUDEP2_L cells. The cluster consists of six miRNAs (miR-106a, miR-18b-, miR-20b, miR-19b-2, miR-92a-2, and miR-363), encoded within about 850 nucleotides, and was previously linked to oncogenesis [58]. Most importantly, the Xq26.2 miRNAs were recently found to be dysregulated by the HPV16-E6/E7 oncoproteins [59]. It is therefore likely that their observed differential expression reflects the doxycycline-induced expression of the HPV16-E6/E7 immortalization system employed in HUDEP cells [39]. The most highly divergent miRNA of this locus was miR-363-3p (HUDEP2_L vs. CD34_L log_2_ FC = 3.51). Overall, chromosomes X, 1 and 19 were the top three most DE miRNA-gene-containing chromosomes, whereas the Y chromosome contained none. The observation was unsurprising, as chromosomes 1, 19, X and 2 are known to carry a disproportionately high number of miRNA genes (about 29% of total human miRNA genes in 15% of human genomic DNA). Moreover, the high number of miRNA genes on the X chromosome and the scarcity or absence of miRNA genes on the Y chromosome are common in all species, constituting an evolutionarily conserved phenomenon [60]. A similar upregulation of the Xq26.2 cluster was observed in HUDEP1_L vs. CD34_L, miR-363-3p being again the most highly divergent of the locus (HUDEP1_L vs. CD34_L log_2_ FC = 5.09) (Appendix A and Appendix A). In line with their HPV16-E6/E7-related expression and with high levels of doxycycline in early HUDEP-1 and HUDEP-2 cultures, the upregulation of the Xq26.2 cluster in HUDEP1_E and HUDEP2_E in comparison to CD34_E cells was even more striking than in late cultures, once more with miR-363-3p as the most upregulated miRNA of the locus (HUDEP1_E vs. CD34_E log_2_ FC = 8.37, HUDEP2_E vs. CD34_E log_2_ FC = 6.69). Two other known oncomiRs, miR-125b-5p and miR-196a-5p [61,62], were also significantly upregulated in HUDEP-2 cells (HUDEP2_L vs. CD34_L log_2_ FC = 4.25, *p* < 1.00 × 10^−300^ and 6.06, *p* = 3.69 × 10^−149^, respectively).

Among the most significantly upregulated miRNAs in CD34_L, we identified miR-126-3p (HUDEP2_L vs. CD34_L log_2_ FC = −1.7), miR-652-3p (HUDEP2_L vs. CD34_L log_2_ FC = −1.0) [63], the HPV16-regulated miR-4707-3p [64], (HUDEP2_L vs. CD34_L log_2_ FC = −12.55), the tumor suppressor miR-143-3p [61] (HUDEP2_L vs. CD34_L log_2_ FC = −8.08) and the miR-221/222 cluster. Both highly sequence-similar miRNAs, miR-221-3p and miR-222-3p, are known to target c-KIT during erythropoiesis and were faintly expressed in HUDEP-2 cells (early and late), with a HUDEP2_L vs. CD34_L log_2_ FC of −7.93 and −9.12, respectively. Their expression was also low in HUDEP-1 cells (early and late), with a HUDEP1_L vs. CD34_L log_2_ FC of −10.42 and −9.08, respectively. Of note, another c-KIT-targeting miRNA, miR-34a-5p [65], showed similar diminished expression in HUDEP-2 compared to hCD34+ cells (HUDEP2_L vs. CD34_L log_2_ FC = −5.27). Likewise, miR-181a-3p, involved in the regulation of erythroid enucleation, was more abundant in primary hCD34+ than HUDEP cells, potentially contributing to the greater enucleation potential of the primary cells (HUDEP2_L and CD34_L log_2_ FC = −5.51 and HUDEP1_L vs. CD34_L log_2_ FC = −4.75). The remaining DE miRNAs between late-stage HUDEP-2 and hCD34+ cells and targeted genes affect pathways related to cell cycle, cellular senescence, oxidative stress-induced senescence, and signaling by Wnt (Appendix A).

### 2.7. Distinct HUDEP-1 and HUDEP-2 miRNA Patterns Reveal Signatures Associated with Hemoglobin Switching

To identify miRNAs implicated in the regulation of fetal-to-adult hemoglobin switching, we comprehensively characterized and compared miRNA profiles of HUDEP1_L (fetal-like) to HUDEP2_L (adult-like) cells. We excluded data obtained from hCD34+ cell cultures as less informative because any comparison of primary to engineered erythroid cells would be affected by additional confounding factors. In total, 274 and 131 known miRNAs were significantly upregulated in HUDEP-1 and HUDEP-2 cells, respectively (Figure 6a and Appendix A), including DE miRNAs linked to hemoglobin switching before (see also Appendix A and Appendix A). The let-7 miRNA family, shown to be upregulated in adult versus fetal erythroblasts in several studies [28,66,67], was the most upregulated family of miRNAs in HUDEP-2 cells. Members of the let-7 miRNA family accounted for 2.10% and 24.52% of all reads mapping to known miRNAs in HUDEP-1 and HUDEP-2, respectively. Several other miRNAs previously put forward as regulators of γ-globin expression through differential expression studies in cord blood vs. adult reticulocytes (e.g., miR-146a and miR-150) [28], in fetal liver vs. bone marrow HSPCs (e.g., miR-98-5p, miR-182-5p and miR-183-5p) [66] or through functional studies (e.g., miR23a-5p, miR-27a-5p, miR-326, and miR-34a) [68,69,70], were also DE between HUDEP1_L vs. HUDEP2_L cells in our study. Two miRNAs, the expression of which has been associated with hydroxyurea-induced γ-globin production, miR-32-5p and miR-340-5p [71], were among the top 40 most significant HUDEP1_L vs. HUDEP2_L DE miRNAs (log_2_ FC = 4.14 and 2.78, respectively). Most of the aforementioned miRNAs are either predicted or experimentally validated to target key regulators of γ-globin production. Several other HUDEP1_L vs. HUDEP2_L DE miRNAs like miR-652-3p, miR-378a-3p, miR-21-5p, and miR-126-3p, have not been previously identified as implicated in γ-globin expression. Three miRNAs predicted to target the open reading frame of γ-globin, miR-96-5p, miR-146a-5p, and let-7a-5p (miR-96-5p also experimentally validated) [28], were significantly upregulated in HUDEP2_L cells (HUDEP1_L vs. HUDEP2_L log_2_ FC = −4.75, −3.55 and −2.99, respectively). Among DE novel miRNAs, novel_mir103 was the most significantly upregulated in HUDEP1_L cells exhibiting a HUDEP1_L vs. HUDEP2_L log_2_ FC of 4.63 (Appendix A). Novel_mir103 was mapped to chr7:156792666_156792743, and interestingly is predicted to target ZBTB7A, a major repressor of human γ-globin gene expression [46].

By mapping each upregulated known miRNA in HUDEP-1 cells to the genomic position of their precursor sequences (Figure 6b), we identified 17 out of a total of 274 as mapping to the frequently imprinted 14q32 locus, previously identified as the genomic locus of a strongly upregulated miRNA cluster in fetal erythroblasts [66]. The most significant miRNA of this locus in our study was miR-342-3p (HUDEP1_L vs. HUDEP2_L log_2_ FC = 1.4). Once more, 5 out of the 6 miRNAs of the Xq26.2 cluster were identified as DE in HUDEP1_L vs. HUDEP2_L, likely reflecting the variability of expression for their shared doxycycline-inducible HPV16-E6/E7 transgene. Chromosomes X and 1 were the top two most miRNA-gene-containing chromosomes for miRNAs upregulated in HUDEP-1 cells, whereas the Y chromosome contained none. By mapping each known miRNA downregulated in HUDEP-1 cells to the genomic position of its precursor sequence, we identified the miR-183 cluster (miR-183, miR-96, and miR-182) located on chromosome 7 [72], and the miR-125a/let-7e cluster on chromosome 19 [73], as significantly downregulated. Chromosome 19 was the topmost miRNA-gene-containing chromosome for miRNAs downregulated in HUDEP-1 cells, whereas chromosomes 13 and Y contained none.

To understand the complex multiple-to-multiple relations between DE miRNAs, target genes, and TFs, we explored TF-miRNA co-regulatory networks using miRNet (Figure 7a,b) [74,75]. The lists of 274 and 131 known miRNAs upregulated in HUDEP1_L and HUDEP2_L, respectively, were used independently as input for network analysis in miRNet and processed along with a separate list of ten TFs known to be involved in γ-globin regulation (BCL11A, ZBTB7A, KLF1, CHD4, KLF10, MYB, SOX6, NFE2, NFYA, TAL1) [76,77,78,79,80]. For five of them (BCL11A, KLF1, MYB, SOX6, TAL1), published gene expression data reveal differential expression between HUDEP-1 and HUDEP-2 cells [39,81]. The analysis was based on experimentally validated miRNA-interaction data collected from the TarBase v8.0 database [82]. The constructed networks, after filtering out of nodes with low degree and betweenness centrality, revealed hub miRNAs and genes as well as important connections between the three (TF-miRNA-gene) interactors. Interestingly, several TFs, including the two major γ-globin repressors, BCL11A and ZBTB7A, were recognized as interactors in both miRNA interaction networks. Among them, ZBTB7A and KLF10 showed the highest degree and betweenness centrality in the network of upregulated miRNAs in HUDEP1_L and HUDEP2_L, respectively. Feedforward loop type-specific interactions, considered to be key components of gene regulatory networks (GRNs) [83,84], were also detected in our networks, shedding more light on particular molecular pathways regulated by these TFs (Appendix A). In an independent analysis of experimentally validated miRNA/target-mRNA interaction networks of BCL11A and ZBTB7A retrieved from miRNet, we found that miRNAs in these networks were overrepresented in the list of HUDEP1_L vs. HUDEP2_L DE miRNAs. More specifically, 9 out of a list of 17 experimentally validated miRNAs that target the major HbF repressor BCL11A according to miRNet, were indicated as DE in HUDEP1-L vs. HUDEP2_L in our study (*p* = 0.0109), whereas the corresponding numbers for ZBTB7A were 35 out of 100 (*p* = 0.0092). We also looked at LIN28B, an RNA-binding protein recently recognized as an upstream regulator of BCL11A [85], and found 17 DE miRNAs out of a list of 41 validated LIN28B-associated miRNAs (*p* = 0.0109) (Appendix A). The above findings advocate a collective role of miRNAs in erythropoiesis and globin expression regulation, where multiple miRNAs act together to fine-tune the expression of important target genes.

To examine the biological interpretation of results further, we performed gene ontology (GO) enrichment analysis [86,87] on target genes of all known and novel HUDEP2_L vs. HUDEP1_L DE miRNAs, to identify GO terms that are significantly overrepresented in our dataset. With numerous overrepresented GO nodes in all three GO categories, *biological process*, *cellular component*, and *molecular function*, the latter with the exemplary nodes *protein binding transcription factor activity* (430 target genes), *antioxidant activity* (42 target genes) and *translational regular activity* (20 target genes) presents nodes of particular potential interest for the differential regulation of globin expression and associated stress factors (Appendix A). In addition, pathway enrichment analysis using Enrichr [88] and WikiPathways 2019 Human [89] identified 106 enriched biological pathways (adjusted *p*-value < 0.05). Several pathways, such as MAPK, PI3K-Akt, RAS, and insulin signaling pathways, have previously been implicated in γ-globin gene regulation [90,91,92,93,94], but others, such as EGF/EGF receptor, VEGF A/VEGF receptor 2, and ErbB signaling pathways, have not been reported as involved in this process before (Table 1).

### 2.8. Evidence Suggests Cross Talk between miRNAs and Long Non-Coding RNAs (lncRNAs)

Accumulating evidence suggests that miRNAs and other classes of ncRNAs interact with each other directly to further modulate the effect of their regulation [95,96]. This multimodal cross talk is only beginning to be unraveled, as new studies, mainly focusing on lncRNA-miRNA interaction, elucidate its regulatory role in biological processes, including erythropoiesis [97,98,99]. We thus conducted in silico analysis of lncRNA-miRNA interactions focusing on BGLT3, an erythroid lncRNA that positively regulates γ-globin gene expression. Lying in an intergenic region downstream of the ^A^γ-globin (*HBG1*) gene, the non-coding *BGLT3* locus and its transcript act separately to increase globin expression through looping to the globin genes and interacting with the mediator of the RNA polymerase II transcription complex, respectively [100]. To date, no BGLT3-miRNA interactions have been reported; however, it is likely that RNA cross-talk exists and modulates the BGLT3-mediated regulation of γ-globin. MicroRNA members of five out of nine miRNA families predicted to interact with BGLT3 according to miRcode [101] were DE in HUDEP1_L vs. HUDEP2_L samples. Three of them, miR-125a-5p, miR-193a-5p and miR-155-5p, were upregulated in HUDEP2_L (HUDEP1_L vs. HUDEP2_L log_2_ FC = −2.7, *p* < 1.00 × 10^−300^ −2.52, *p* = 1.24 × 10^−178^ and −1.09, *p* < 1.00 × 10^−300^, respectively) and two, miR-142-3p and miR-138-5p, upregulated in HUDEP1_L cells (HUDEP1_L vs. HUDEP2_L log_2_ FC = 2.42, *p* < 1.00 × 10^−300^ and 1.21, *p* = 1.29 × 10^−4^, respectively). Interestingly, miR-125a-5p and miR-142-3p were among the top 50 most significant HUDEP1_L vs. HUDEP2_L DE miRNAs. Of note, HBBP1 (NR_001589), another lncRNA recently identified as related to high γ-globin levels [102], is predicted to interact with five HUDEP1_L vs. HUDEP2_L DE miRNAs, including miR-18b-5p (HUDEP1_L vs. HUDEP2_L log_2_ FC = 2.99), which was among the top 20 DE miRNAs based on *p*-value.

### 2.9. Reverse-Transcription Quantitative PCR (RT-qPCR) Validates Small RNA Sequencing Results

Five miRNAs, miR-451a, miR-182-5p, miR-146a-5p, miR-221-3p and miR-222-3p, showing significant expression variation between sample groups, were selected to verify the sequencing results by RT-qPCR in the same samples. The expression patterns of the five miRNAs were quantified by stem-loop RT-qPCR and plotted along with the corresponding sequencing data for comparison (Figure 8). MiR-451a was verified as associated with erythroid maturation, showing significant upregulation in hCD34+ (CD34+_L vs. CD34+_E log_2_ FC = 5.64, *p* = 2.91 × 10^−4^) and HUDEP-2 cells (HUDEP-2_L vs. HUDEP-2_E log_2_ FC = 3.87, *p* = 2.68 × 10^−3^), while miR-182-5p was significantly upregulated only during erythroid maturation of hCD34+ (CD34+_L vs. CD34+_E log_2_ FC = 6.82, *p* = 8.05 × 10^−3^). We also confirmed the minimal expression of miR-221-3p and miR-222-3p in HUDEP cells as well as the differential expression of miR-146a-5p between fetal-like HUDEP-1 and adult-like HUDEP-2 cells (HUDEP1_L vs. HUDEP2_L log_2_ FC = −3.15, *p* = 1.29 × 10^−2^ and HUDEP1_E vs. HUDEP2_E log_2_ FC = −5.09, *p* = 4.83 × 10^−5^).

Overall, all five miRNAs identified as DE by sequencing and shortlisted for confirmation by RT-qPCR showed the expected trend and similar magnitude of dynamic expression in RT-qPCR analysis. The high concordance between the two methods as calculated by Pearson correlation coefficient analysis (*r* = 0.8369, *p* < 0.0001) validates our general results obtained by small RNA sequencing.

## 3. Discussion

With this study, we provide the first comprehensive analysis of small RNAs expressed during erythroid lineage progression in HUDEP cell lines, along with a parallel analysis in adult hCD34+ cells.

In summary, miRNA expression analysis revealed a pattern of time-related changes during erythroid differentiation in both primary hCD34+ cells and HUDEP erythroid cell lines. In the process, more miRNAs were downregulated than upregulated, yet terminal stages of erythropoiesis were marked by a sharp increase in the expression of several lineage-specific miRNAs. Same-stage hCD34+ and HUDEP-2 cells (adult-like erythroid cells) showed evident clustering and clear separation from HUDEP-1 cells (fetal-like erythroid cells). Differential expression analysis in HUDEP1_L vs. HUDEP2_L cells suggested several miRNAs and implicated their putative target genes and related pathways in γ-globin regulation. Corresponding TF-miRNA co-regulatory networks readily incorporated recognized major factors involved in γ-globin regulation, including individual genes targeted simultaneously by multiple miRNAs or multiple genes targeted by the same miRNAs.

To elaborate on the “multiple-to-multiple” pattern of miRNA interaction, we explored miRNA/target-mRNA interaction networks of key regulators of hemoglobin switching (BCL11A and ZBTB7A) and found the corresponding miRNAs significantly enriched in the list of DE miRNAs between HUDEP-1 and HUDEP-2 cells. As our understanding of miRNA biology advances, it becomes clearer that multiple miRNAs act in a combinatorial and synergistic manner on the same target or pathway to regulate developmental events, such as hemoglobin switching. Groups of miRNAs that are “coordinately” increased and/or decreased at a certain cell state are likely to participate in interwoven GRNs, where recurrent motifs of feedback and feedforward loops exist between TFs, miRNAs, and their shared target genes [83,103,104]. Recent studies analyzing high-throughput data for both miRNAs and mRNAs have noted the prevalence of several miRNA/target-mRNA pairs with positively correlated expression patterns. These observations contradict the traditional notion of negatively correlated miRNA/target-mRNA pairs, which is based on an exclusively repressive mode of action of miRNAs but can be explained if perceived in the context of GRNs. In line with this, several up- and downregulated miRNAs in HUDEP-1_L in comparison to HUDEP-2_L cells were found to interact with some of the known γ-globin repressors in our constructed miRNA/target-mRNA networks. The identification of those co-expressed miRNAs that jointly regulate genes of interest (either cooperatively or antagonistically) in HbF expression pathways partly explains why most studies focusing on manipulating the expression of single miRNAs have failed to reactivate HbF substantially [27,28,69,105]. Mild co-manipulation of more than one miRNA may have a more pronounced phenotypic effect than targeting single miRNAs while minimizing at the same time off-target effects [106]. In this vein and beyond clear correlation with established mRNA expression patterns, integration of our data with comprehensive mRNA profiling will allow deeper insights into the networks concerned, as would multiplex inactivation of identified miRNA networks to validate their putative action on individual mRNA targets.

Adding to the complexity of GRNs, a new layer of regulation of biological processes was unraveled with the discovery of lncRNAs and the understanding of their role as key mediators in ceRNA activity. Recent theoretical and experimental studies have shed light on RNA-RNA “communication,” reporting several modes of interaction: miR-triggered RNA decay, competition between miRNAs and lncRNAs for the same mRNA target, miRNA generation from lncRNAs, and lncRNAs acting as decoys for miRNAs [107]. Computational clues in our study provide preliminary evidence that for several miRNAs put forward as implicated in globin regulation here, two lncRNAs, BGLT3 and HBBP1, previously shown to be associated with high γ-globin levels, besides regulating the expression of target genes independently may act as miRNA binding partners to further modulate their effect. The versatility of lncRNA interactions and function, however, calls for dedicated functional studies to determine the nature of such interactions and draw firm conclusions as to their effect.

Our study is the first to determine miRNA expression profiles of HUDEP cells at early and late stages of erythroid differentiation and directly compare HUDEP cells to same-stage primary HSPC-derived erythroid cells. Given the now-widespread use of HUDEP-2 cells as a model of human terminal erythropoiesis [34,35,36,37,38], recognizing existing differences between them and their native counterparts is important for the design and interpretation of experimental studies based on HUDEP cell lines. This is particularly critical for the development of targeted therapies, as manipulated cellular pathways may respond differently in engineered cell lines compared to their tissue of origin [108]. We have highlighted specific aspects of HUDEP cells that need to be taken into account during the interpretation of functional studies. These include the inherent elevation of HPV16-E6/E7-related pathway for both, HUDEP-1 and HUDEP-2 cells, which is related to the immortalization process and not representative of observations in primary cells. Related to this point, differences in miRNA expression from the Xq26.2 cluster were also detected between both HUDEP cells and likely reflect differential transgene expression, short half-life and/or varying metabolization of doxycycline, and thus are probably not of biological significance for natural erythropoiesis and globin switching.

Weighed against greater fidelity of individual primary samples for observations in vivo, the use in the present study of cell lines HUDEP-1 and HUDEP-2 for the identification of γ-globin-related miRNA signatures offers the substantial benefit of directly comparing co-derived fetal-like (HUDEP-1) cells to adult-like (HUDEP-2) cells of common tissue origin, specifically human umbilical cord blood, under the same culture conditions and without the biological variation typical of primary samples. In contrast with previous studies comparing erythroblasts derived from different tissues (fetal liver or cord blood vs. adult BM or PB) [28,66,67], the common tissue-specific background expression in our cord blood-derived erythroblasts renders any identified differences more likely to be relevant to γ-globin regulation. Likewise, a modest sample size and its impact on statistical sensitivity and specificity in our study are potentially offset by vastly reduced intra-group variation for cell lines, a view vindicated by consistent agreement of our significant observations with existing literature, where there are corresponding data. Moreover, the BGISEQ-500 sequencing system utilized here has been reported as superior to the commonly used Illumina HiSeq system for the detection of novel miRNAs, by showing a more even coverage of rare miRNA species, which further strengthens the validity of our novel data [109]. Taken together, these factors, in turn, lend credence to our novel observations, such as for novel miRNAs (e.g., novel_mir1 and novel_mir103) and novel or extended GRNs, including miRNAs and pathways newly implicated in erythropoiesis and γ-globin gene regulation, which are likely of biological relevance also in primary cells. Our work thus puts forward specific candidates and hypotheses for validation in functional studies, such as by combinatorial knock-down or editing of co-regulated and potentially collaborating miRNAs.

In conclusion, this study provides important insights into miRNA regulatory mechanisms in erythropoiesis and globin expression, highlighting co-regulation of miRNAs as central to their natural function. MiRNAs, genes, and pathways identified through this study may be explored as potential targets for the development of novel therapies for β-hemoglobinopathies and other disorders of erythropoiesis. Moreover, this study demonstrates the similarity of the commonly used HUDEP-2 cell line to primary erythroid cells, also at the level of miRNAs, and vindicates the use of HUDEP-2 and HUDEP-1 lines for corresponding functional studies.

## 4. Materials and Methods

### 4.1. Culture of Human Umbilical Cord Blood-Derived Erythroid Progenitor (HUDEP) Cells

HUDEP cells were expanded in StemSpan SFEM II (Stem Cell Technologies, Vancouver, BC, Canada) supplemented with 1 µM Dexamethasone (Sigma-Aldrich, Munich, Germany), 5 μg/mL Doxycycline (Clontech Laboratories, Mountain View, CA, USA), 100 ng/mL Recombinant Human Stem Cell Factor (hSCF) (PeproTech, Rocky Hill, CT, USA), 3 IU/mL Epoetin alpha (EPO) (Binocrit 4000 IU/0.4 mL, Sandoz GmbH, Kundl, Austria), and 2× Penicillin-Streptomycin (Thermo Fisher Scientific, Waltham, MA, USA) at concentrations below 0.5 × 10^6^ cells/mL. The culture medium was changed twice a week and doxycycline was replenished every other day to retain expression of the HPV16-E6/E7 immortalization system. To induce erythroid differentiation, cells were cultured in a three-phase erythroid differentiation culture system previously described [36]. Briefly, cells were cultured in Iscove’s Modified Dulbecco’s Medium (Thermo Fisher Scientific, Waltham, MA, USA) containing 5% Human Serum from plasma male AB (Sigma-Aldrich, Munich, Germany), 2× Penicillin-Streptomycin (Thermo Fisher Scientific, Waltham, MA, USA), 1% L-Glutamine (Thermo Fisher Scientific, Waltham, MA, USA), 200 µg/mL Holo-Transferrin (Sigma-Aldrich, Munich, Germany), 2 IU/mL Heparin (Sigma-Aldrich, Munich, Germany), 10 ng/mL Insulin (Sigma-Aldrich, Munich, Germany), and 3 IU/mL EPO (Binocrit 4000 IU/0.4 mL, Sandoz GmbH, Kundl, Austria). In phase I (days 1–4), the culture medium was supplemented with 100 ng/mL hSCF (PeproTech, Rocky Hill, NJ USA). In phase II (days 5–7), hSCF was withdrawn from culture. In phase III (days 8 and 9), doxycycline was also removed from the culture medium. Cells in differentiation were maintained at 1 × 10^6^ cells/mL. Cell counting was performed manually using a hemocytometer, and cell viability was measured using Trypan Blue Solution, 0.4% (Thermo Fisher Scientific, Waltham, MA, USA).

### 4.2. Isolation and Culture of PB-Derived hCD34+ HSPCs

hCD34+ cells were isolated from PB of healthy individuals by magnetic-activated cell sorting, after mononuclear cell isolation using a density gradient medium and according to a published protocol (Protocol C) [110]. Briefly, PB was collected in K2EDTA tubes (Greiner Bio-One, Kremsmünster, Austria) and mixed with 1.5 blood-volumes of Dulbecco’s Phosphate-Buffered Saline (PBS) without calcium chloride and magnesium chloride (Sigma-Aldrich, Munich, Germany), and then carefully overlaid on top of 1.25 blood-volumes of Lymphoprep medium (Axis-Shield, Dundee, UK). After centrifuging at 900× *g* for 40 min without brake, the interphase, consisting of mononuclear cells, was harvested, washed 3 times in PBS, twice by centrifugation at 300× *g* for 5 min and once at 200× *g* for 5 min. The cell pellet was then resuspended in 600 μL ice-cold 1% bovine serum albumin (BSA) (Sigma-Aldrich, Munich, Germany) in PBS and incubated for 15 min with 100 μL CD34 MicroBead from CD34+ MicroBead Kit, human (Miltenyi Biotec, Bergisch Gladbach, Germany), on a rocking shaker at 4 °C. After two washes with ice-cold beading buffer (BB) (0.5% BSA, 2 mM EDTA in PBS) and centrifugation at 300× *g* for 5 min at 4 °C, the cell pellet was resuspended in 1 mL BB (without EDTA). Magnetic cell separation was carried out using LS columns placed on the QuadroMACS Separator on a MACS MultiStand (all from Miltenyi Biotec, Bergisch Gladbach, Germany). LC column was equilibrated with 5 mL BB, and then cells were loaded onto the column in two consecutive steps with an intervening BB column washing step. Cells were eluted from the column with 5 mL BB (by flushing with the plunger) and for the second round of selection were loaded onto a new equilibrated column to enrich further for hCD34+ cells. Cells were again eluted with 5 mL BB and collected by centrifugation at 300× *g* for 5 min. Isolated cells were then expanded in StemSpan SFEM II (Stem Cell Technologies, Vancouver, Canada) supplemented with 1 µM Dexamethasone (Sigma-Aldrich, Munich, Germany), 2 IU/mL EPO (Binocrit 4000 IU/0.4 mL, Sandoz GmbH, Kundl, Austria), 1% StemSpan CC100 (Stem Cell Technologies, Vancouver, Canada) and 1× Penicillin-Streptomycin (Thermo Fisher Scientific, Waltham, MA, USA) at concentrations below 0.5 × 10^6^ cells/mL for ten days. The medium was changed twice a week. For the induction of erythroid differentiation, cells were cultured in Minimum Essential Medium Eagle, Alpha modification (Sigma-Aldrich, Munich, Germany) supplemented with 30% Defined Fetal Bovine Serum (Hyclone, Logan, Utah, USA), 10^−5^ M 2-Mercaptoethanol (Sigma-Aldrich, Munich, Germany), 10 IU/mL EPO (Binocrit 4000 IU/0.4 mL, Sandoz GmbH, Kundl, Austria) 10 ng/mL hSCF (PeproTech, Rocky Hill, USA) and 1× Penicillin-Streptomycin (Thermo Fisher Scientific, Waltham, MA, USA) at a concentration of 0.5–1 × 10^6^/mL for six days. Again, cell counting was performed manually using a hemocytometer, and cell viability was measured using Trypan Blue Solution, 0.4% (Thermo Fisher Scientific, Waltham, MA, USA).

### 4.3. Cytocentrifugation and Microscopy

For determining cell morphology, 200 μL of cell suspension containing 1 × 10^5^ cells were used for cell preparations on slides, using the Tharmac Cellspin II cytocentrifuge with an EASY-rotor (Tharmac, Wiesbaden, Germany). The slides were fixed in ice-cold methanol for 4 min, immersed in 1.5% o-Dianisidine solution (Sigma-Aldrich, Munich, Germany) for 2 min and then in freshly prepared H_2_O_2_/ethanol solution (50% ethanol, 1.5% H_2_O_2_ in dH_2_O) for another 2 min. A standard May–Grünwald/Giemsa (Fluka, Munich, Germany) staining procedure was then followed, after which slides were air-dried, mounted, and sealed with a coverslip. Slide images were acquired using an IX73P1F inverted microscope, LED illumination, a 40× lens, and CellSens 1.7 imaging software (all Olympus Corporation, Shinjuku City, Tokyo, Japan). Four or five images per sample were obtained by using an averaged acquisition of seven frames, and cell morphology was evaluated blinded to the cell type and phase.

### 4.4. Flow Cytometry and Analysis

1 × 10^5^ cells were used for each staining, including a non-stained sample for each set of measurements. Briefly, cells were washed twice in PBS (Sigma-Aldrich, Munich, Germany) and incubated for 10 min at 4 °C in 40 μL blocking buffer, composed of 39 μL 1% Bovine Serum Albumin (BSA) (Sigma-Aldrich, Munich, Germany) in PBS (Sigma-Aldrich, Munich, Germany) and 1 μL FcR human blocking reagent (Miltenyi, Bergisch Gladbach, Germany). Cells were then incubated with conjugated antibodies (PE anti-human CD235a (Glycophorin A) Antibody (clone HIR2), FITC anti-human CD71 Antibody (clone CY1G4), FITC anti-mouse/human CD44 Antibody (clone IM7), FITC anti-human CD49d Antibody (clone 9F10), all from Biolegend, San Diego, CA, USA, and at concentrations suggested by the manufacturer) for 30 min at 4 °C in the dark and analyzed using a CyFlow Cube 8 6-channel instrument (Sysmex Partec, Münster, Germany). For cell death assessment, cells were incubated for 10 min at 4 °C in the dark with 7-AAD Viability Staining Solution (Biolegend, San Diego, CA, USA) according to the manufacturer’s instructions. A minimum of 0.5 × 10^5^ cellular events was recorded and data were analyzed and visualized using FCS Express 4 (*DeNovo Software*).

### 4.5. Globin Chain Analysis by RP-HPLC

The pellet of 1 × 10^6^ differentiated (_L) cells was washed with PBS and resuspended in 50 μL of HPLC-grade water before two rounds of freezing/thawing. After centrifugation at 16,000× *g* for 10 min at 4 °C, the supernatant was transferred to HPLC vials (Altmann Analytik, Munich, Germany). An LC-20AD chromatographic system (Shimadzu, Kyoto, Kyoto, Japan) and an Aeris Widepore C18 column (Phenomenex, Torrance, CA, USA) were used to separate peptides based on their hydrophobicity and on an increasing linear gradient of acetonitrile with 0.1% trifluoroacetic acid against 0.1% trifluoroacetic acid/0.033% sodium hydroxide for elution from the column, as previously published [111]. 25‒30 μL of protein extract were injected per analysis. Heme and globin chains were eluted from the column at different retention times and detected as absorbance peaks, the area of which was used to determine the relative quantities of globin chains in samples.

### 4.6. DNA Nanoball (DNB) Small RNA Sequencing and Analysis

After RNA extraction, small RNAs were size-selected (18–30 nucleotides) using polyacrylamide gel electrophoresis. A total of 18 small RNA libraries were constructed comprising three biological replicates for each of the following samples and time points: CD34_E, HUDEP1_E, HUDEP2_E, CD34_L, HUDEP1_L, and HUDEP2_L. Sequencing was performed using DNB sequencing technology on a BGI-SEQ500 sequencing platform generating ~ 40 × 10^6^ 50-bp single-end reads per sample. Sequencing data were submitted to NCBI GEO with accession ID GSE165011. Before data analysis, raw data were filtered to remove low-quality reads, adaptors, and other contaminants, and then clean reads were mapped to the reference genome and to other ncRNA databases, such as miRBase [112] and Rfam 12.0 [113] by using Bowtie 2 [114] and CMsearch programs [115]. Micro-RNA counts were normalized to transcripts per million (TPM) [116] and DE analysis was performed using DEGseq for replicate samples [47]. DE miRNAs were defined as miRNAs with FDR < 0.001 and a FC of ≥2 up or down. Of note and in disambiguation of varying usage by authors and service providers, DE of sample A vs. sample B was expressed throughout as levels for sample A relative to levels for sample B. Throughout this study, “known” miRNAs and piRNAs are defined as those included in miRBase database [112] and piRNABank [117], respectively, whereas, “novel” miRNAs and piRNAs are those predicted by miRDeep2 [118] and Piano [119], respectively. PCA of the normalized quantification (as TPM) of 458 miRNAs with TPM readout > 0.02 for ≥3 samples and with a relative SD < 1.0 per biological replicate trio of samples, was performed in ClustVis [120]. Singular Value Decomposition with imputation was used to calculate principal components and the two major components selected for display. Hierarchical clustering was performed using the pheatmap function [121] and was based on the intersection of miRNAs DE for all three comparisons shown per panel. For target gene prediction, the minimum free energy (MFE) threshold was set to −20 kcal/mol and the intersection of the target genes of RNAhybrid [122], miRanda [123], and Targetscan [124] was chosen as the final dataset used in downstream analyses. For a visual exploration of TF-miRNA co-regulatory networks, we used miRNet and TarBase v8.0 databases [75,82]. To identify hub genes with high connectivity in networks and to illustrate the “multiple-to-multiple” concept of miRNA action, we applied filtering based on degree centrality (cutoff 1), betweenness centrality (cutoff 1), and “shortest path” in miRNet. GO and pathway enrichment analyses were performed for DE target genes using WEGO software [125] and Reactome [54] or Wikipathways in Enrichr [88,89], respectively. To look at lncRNA genes and predicted lncRNA-miRNA interactions, we used miRcode [101].

### 4.7. Reverse-Transcription Quantitative PCR (RT-qPCR)

Quantitative PCR using TaqMan MicroRNA Assay (Applied Biosystems, Foster City, CA, USA) was conducted to validate the sequencing results. 2–4 × 10^6^ cells were lysed in 1 mL of TriFast (Peqlab/VWR, Erlangen, Germany), and miRNAs were extracted using PureLink™ miRNA Isolation Kit (Invitrogen, Carlsbad, CA, USA), according to the manufacturer’s instructions. cDNA was generated using TaqMan MicroRNA Reverse Transcription Kit (Applied Biosystems, Foster City, CA, USA) and a miR-specific stem-looped RT Primer (TaqMan MicroRNA Assay, Applied Biosystems, Foster City, CA, USA) according to the manufacturer’s instructions. RT-qPCR reactions were performed in 96-well plates on a QuantStudio 7 Flex Real-Time System (Applied Biosystems, Foster City, CA, USA). The reactions were prepared in a volume of 10 μL containing 0.67 μL cDNA, 5 μL TaqMan Universal PCR Master Mix (Applied Biosystems, Foster City, CA, USA), and 0.5 μL of miR-specific TaqMan MicroRNA Assay (a mix of miR-specific forward PCR Primer, miR-specific reverse PCR Primer, and miR-specific TaqMan MGB probe) (Applied Biosystems, Foster City, CA, USA). Reaction conditions were as follows: an initial enzyme activation step for 10 min at 95 °C, followed by 40 cycles of denaturation for 15 s at 95 °C and elongation for 60 s at 60 °C. All reactions were performed in triplicates and a non-template control sample was always included as a negative control. The expression of five miRNAs was analyzed and normalized against miR-93-5p. Three commonly applied housekeeping small RNAs were initially selected and evaluated as candidate internal controls: RNU48, miR-26a-5p, and miR-93-5p [126,127,128,129]. Among them, miR-93-5p exhibited the best stability (smallest CT variation) and was hence identified as the most suitable internal normalizer for our dataset. In accordance with this, miR-93-5p was the gene with the lowest coefficient of variation identified by small RNA sequencing in our study (Appendix A). Exemplary analysis of efficiencies for miR-93-5p and miR-451a confirmed equal amplification efficiencies (similar slope values at −3.73 (R^2^ = 0.99) and −3.55 (R^2^ = 0.99), respectively, *n* = 3) across the entire sample dilution range. Determination of CTs was performed using the QuantStudio 7 Software v1.7.1 (Applied Biosystems, Foster City, CA, USA), and raw CT data were transformed into relative quantities by the Δ∆CT method [130].

### 4.8. Statistical Analysis

After data processing in Excel (Office 2013, Microsoft Corp, Redmond, WA, USA), enrichment analyses for miRNA/target-mRNA pairs (based on miRNet-validated represented and DE miRNAs for each target and the input list) were performed as hypergeometric distribution analysis in Excel, and all other statistical analyses were performed in Prism 8.0 (GraphPad Software Inc., La Jolla, CA, USA). The D’Agostino-Pearson normality test was used to determine sample distribution and the appropriate test for non-directional group comparisons. One-way ANOVA with Dunnett’s or Tukey’s multiple comparison test, as appropriate, was used to compare normally distributed data, for others the non-parametric Kruskal–Wallis test with Dunn’s multiple comparison test. Summary statistics are given as geometric mean for fold-changes and as the arithmetic mean ± SD of the population mean for other data. To measure the strength of association between sequencing and RT-qPCR data, we determined the Pearson correlation coefficient r and its corresponding *p*-value.

## Figures and Tables

**Figure 1 ijms-22-03626-f001:**
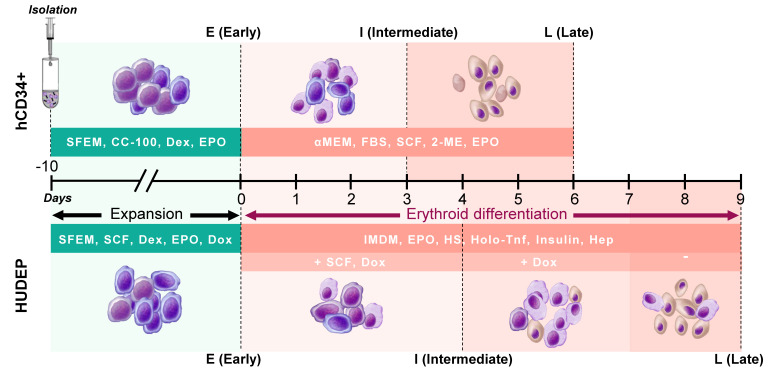
Experimental design. A schematic diagram showing cell-type-specific culture stages, conditions, and timelines. hCD34+: primary human peripheral-blood-derived CD34+, HUDEP: cell lines HUDEP-1 and HUDEP-2, SFEM: Stem Span SFEM II, αMEM: Minimum Essential Medium Eagle with alpha modification, IMDM: Iscove’s Modified Dulbecco’s Medium, FBS: fetal bovine serum, SCF: stem cell factor, CC-100: StemSpan CC100, Dexa: dexamethasone, Dox: doxycycline, EPO: erythropoietin, HS: human serum, Holo-Tnf: holo-transferrin, Hep: heparin, 2-ME: 2-mercaptoethanol.

**Figure 2 ijms-22-03626-f002:**
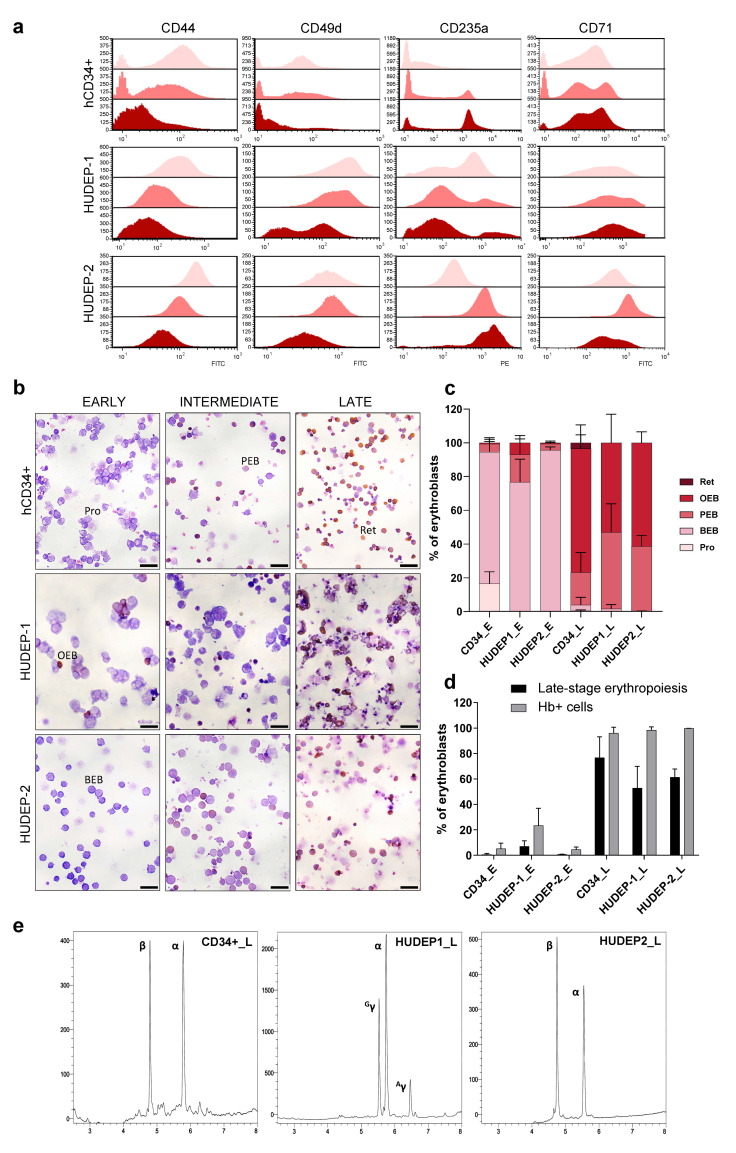
Assessment of erythroid differentiation in cell culture systems. (**a**) Flow cytometric analysis of indicated surface markers (CD44, CD49d, CD235a, and CD71) during erythroid differentiation. Colors in stacked overlaid histograms represent the three stages of erythroid differentiation: _E (pink), _I (light red), and _L (dark red). The ordinate represents the number of cells displaying the fluorescent intensity given by the abscissa. (**b**) Representative images of o-Dianisidine and May–Grünwald/Giemsa staining of cytocentrifugation samples prepared at _E, _I, and _L stages of erythroid differentiation in hCD34+, HUDEP-1, and HUDEP-2 cells. Scale bars: 25 μm. (**c**) Average percentages of erythroid subpopulations across all samples according to differential counting of cells. (**d**) Average percentages of cells in late-stage erythropoiesis (orthochromatophilic and reticulocytes) (black bars) and average percentages of hemoglobin-producing cells (o-Dianisidine positive) (grey bars). (**e**) Reversed-phase high-performance liquid chromatography chromatograms of globin expression in _L cultures. Pro: proerythroblast, BEB: basophilic erythroblast, PEB: polychromatophilic erythroblast, OEB: orthochromatophilic erythroblast, Ret: reticulocyte.

**Figure 3 ijms-22-03626-f003:**
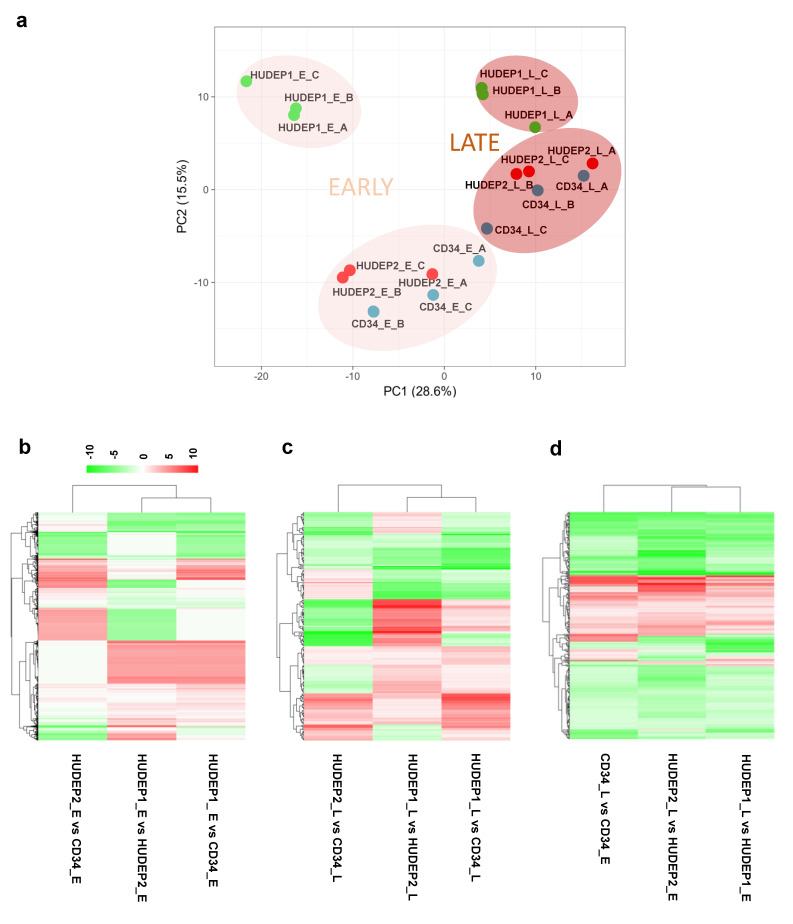
Comparative analysis of miRNA expression in primary hCD34+ and HUDEP cells during erythroid differentiation. (**a**) Principal component (PC) analysis of known and novel miRNA expression. Clustering of samples by differentiation stage is denoted by colors (pale pink: early, dark pink: late). (**b**,**c**) Hierarchical clustering of all known and novel, consistently differentially expressed (DE) miRNAs for early-stage (**b**) and late-stage (**c**) comparisons between cell types. (**d**) Hierarchical clustering of all known and novel, consistently DE miRNAs for erythroid differentiation (_L vs. _E) of all three cell types. For (**b**–**d**), the X-axis represents pairwise comparisons and the Y-axis represents DE miRNAs; for (**d**), the child node was rotated post-analysis to place HUDEP-2 cells in the center of the heat map. The colors indicate the fold change, with red showing upregulation, and green showing downregulation.

**Figure 4 ijms-22-03626-f004:**
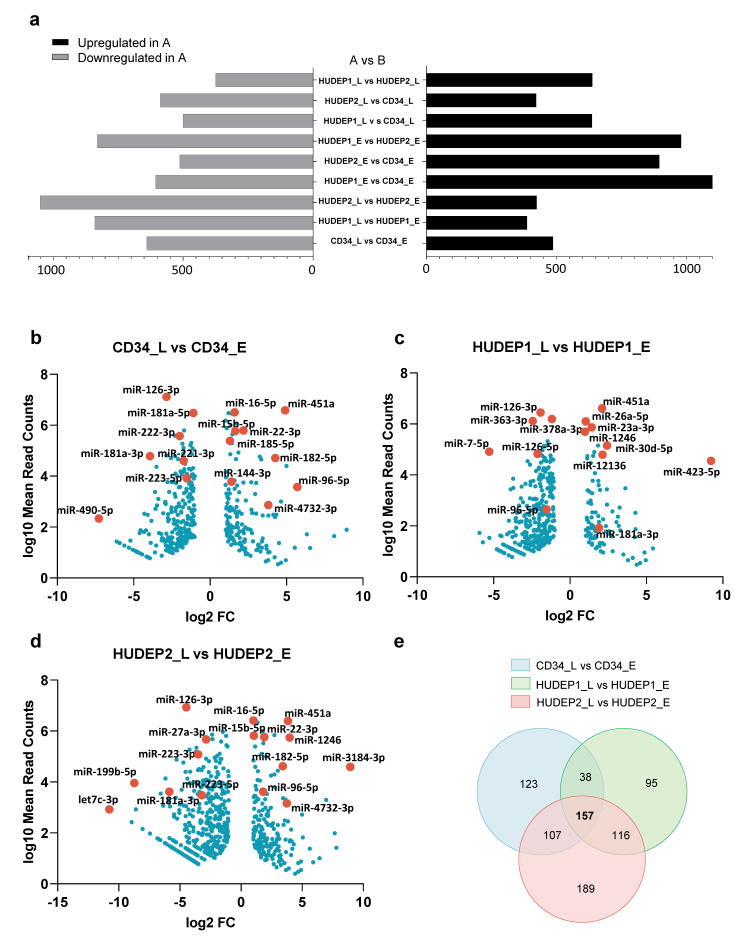
Differential expression analysis of _L vs. _E stages for known and novel miRNAs. DEGseq reveals an overall common erythroid differentiation signature but highlights discrepancies between samples. (**a**) Horizontal bar graph showing pairwise differential expression analysis of all known and novel miRNAs. The X-axis represents numbers of differentially expressed (DE) miRNAs and the Y-axis pairwise comparisons. Late- vs. early-stage expression reveals a higher number of downregulated than upregulated miRNAs during erythroid differentiation. (**b**–**d**) DE known miRNAs in CD34_L vs. CD34_E, HUDEP1_L vs. HUDEP1_E, and HUDEP2_L vs. HUDEP2_E, respectively. Fold changes were plotted as the log_2_ FC of miRNA expression in samples against the log_10_ mean normalized miRNA counts calculated by differential expression analysis. To aid legibility, only a selection of miRNAs of interest and DE miRNAs with high fold change or high levels of expression are labeled (see Appendix A for a full list). (**e**) Venn diagram showing overlapping known DE miRNAs for all _L vs. _E comparisons.

**Figure 5 ijms-22-03626-f005:**
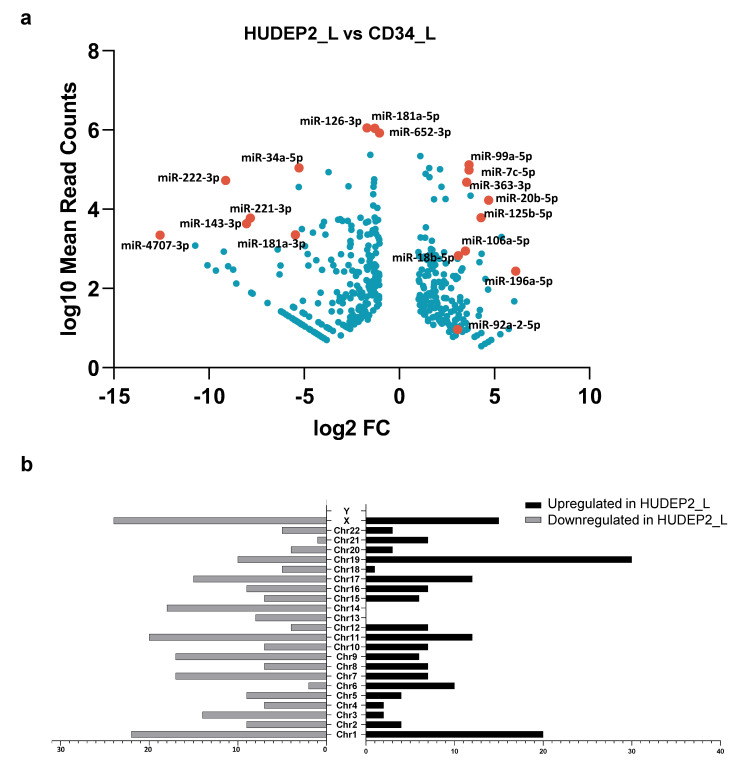
Differential expression analysis in HUDEP2_L vs. CD34_L for known miRNAs. DEGseq highlights differences between primary cells and a prevalent cell model. (**a**) Differentially expressed (DE) known miRNAs in HUDEP2_L vs. CD34_L. Fold changes were plotted as the log_2_ FC of miRNA expression in samples against the log_10_ mean normalized miRNA counts calculated by differential expression analysis to aid legibility, only a selection of miRNAs of interest and DE miRNAs with high fold change or high levels of expression are labeled (see Appendix A for a full list). (**b**) The number of HUDEP2_L vs. CD34_L DE known miRNAs located in different chromosomes.

**Figure 6 ijms-22-03626-f006:**
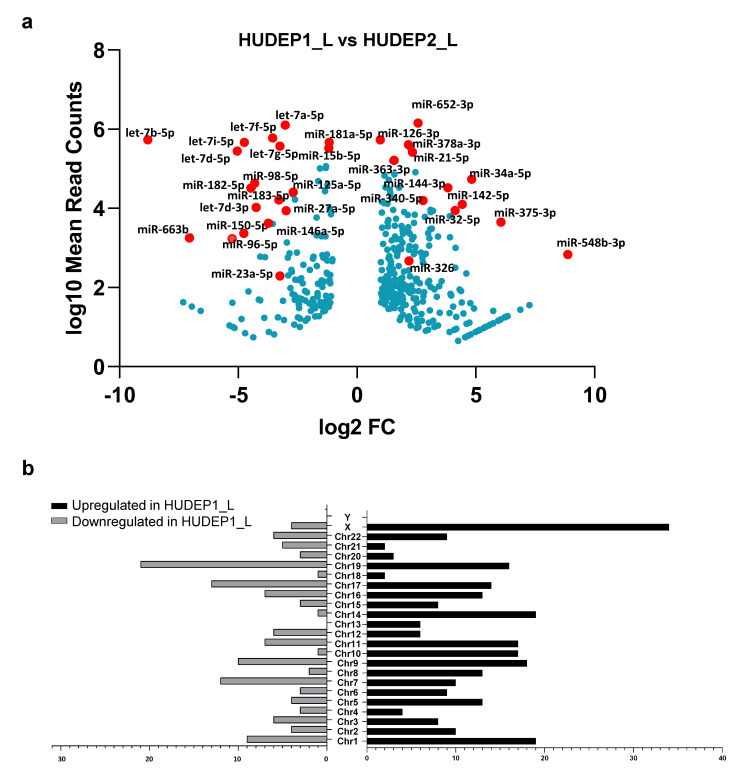
Differential expression analysis in HUDEP1_L vs. HUDEP2_L for known miRNAs. DEGseq provides pointers to miRNAs and related pathways mechanistically linked to globin switching. (**a**) Differentially expressed (DE) miRNAs in HUDEP1_L vs. HUDEP2_L. Fold changes were plotted as the log_2_ FC of miRNA expression in samples against the log_10_ mean normalized miRNA counts calculated by differential expression analysis. To aid legibility, only a selection of miRNAs of interest and DE miRNAs with high fold change or high levels of expression are labeled (see Appendix A for a full list). (**b**) The number of HUDEP1_L vs. HUDEP2_L DE known miRNAs located in different chromosomes.

**Figure 7 ijms-22-03626-f007:**
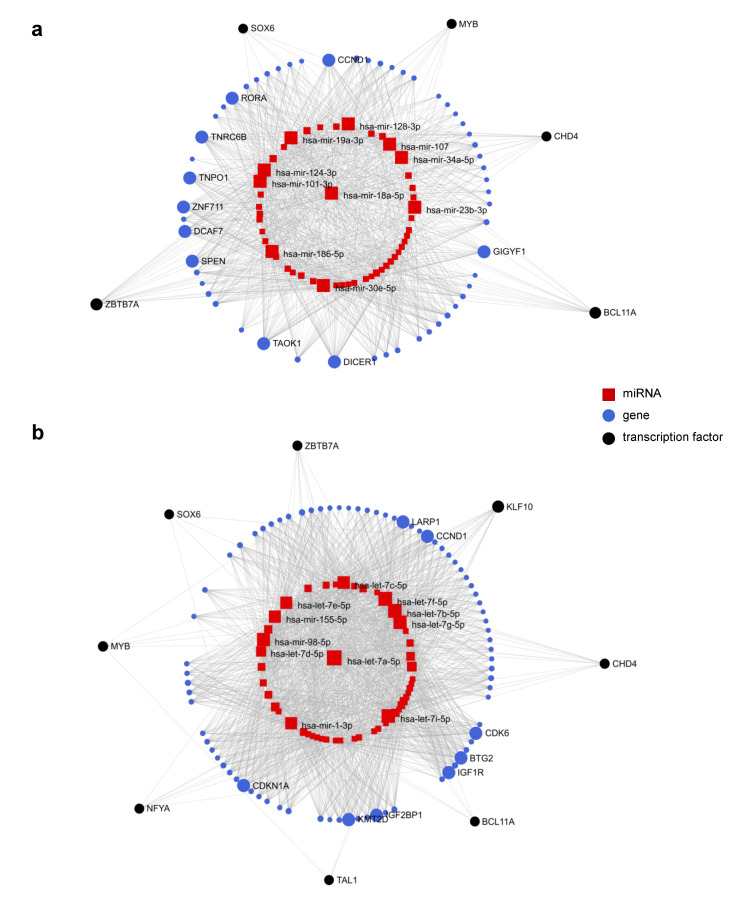
TF-miRNA co-regulatory networks illustrating the “multiple-to-multiple” pattern of miRNA interaction. Networks were extracted from miRNet using key erythroid transcription factors and DE miRNAs as input (accessed 12/2020). Analyses were performed in line with miRNet recommendations to limit network size (<2000 nodes), i.e., the “Data Filter” function was used, based on the network topology measures—degreed, betweenness, and shortest path. In this context, the degree of a node is the total number of connections it has to other nodes, and high-degree nodes are considered important “hubs” in a network (set to the default 1.0 here). The betweenness measures the number of shortest paths going through a node, and nodes with higher betweenness are important interactors in a network (set to 1.0 here). The shortest path filter helps reduce the complexity of a network by keeping only one, the shortest, path between hub nodes, i.e., in the presence of multiple paths connecting two nodes, only the shortest path will be retained (applied here). In the figure, node size indicates betweenness centrality, with large symbols highlighting hub nodes. The components shown are (from inside out) input miRNAs mapped to a network, miRNet-identified target genes, and input TFs mapped to a network, as indicated with corresponding symbols. (**a**) TF-miRNA co-regulatory network of known miRNAs upregulated in HUDEP1_L. (**b**) TF-miRNA co-regulatory network of known miRNAs upregulated in HUDEP2_L.

**Figure 8 ijms-22-03626-f008:**
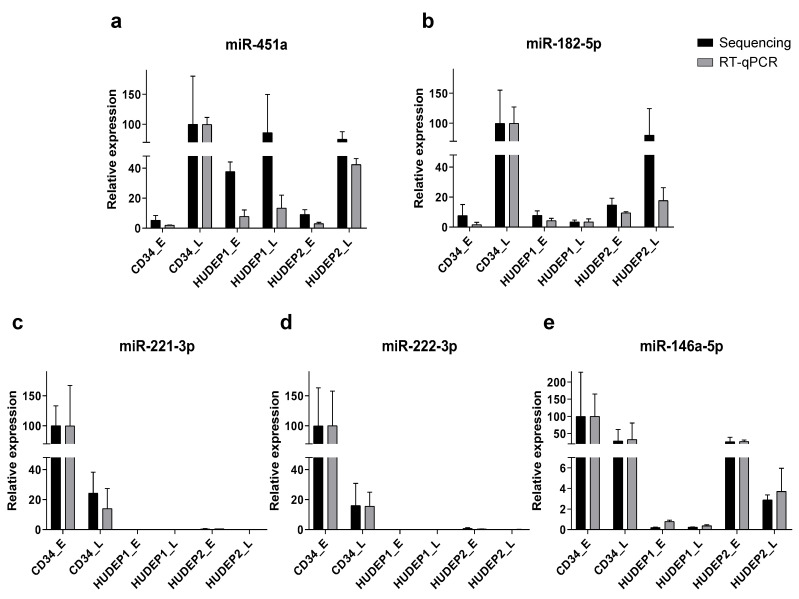
Validation of sequencing data with RT-qPCR analysis of selected miRNAs. (**a**–**e**) The expression of five miRNAs (miRNA-451a, miRNA-182-5p, miR-221-3p, miR-222-3p, and 146a-5p) was analyzed by RT-qPCR and normalized using miRNA-93-5p as an endogenous control for RNA input. All RT-qPCR experiments were carried out in triplicates. The miRNA expression is presented as mean ± standard deviation (SD), normalized to the highest sample readout.

**Table 1 ijms-22-03626-t001:** Top enriched pathways in HUDEP2_L vs. HUDEP1_L DE miRNAs ranked by combined score value.

Pathways	*p*-Value	Adjusted *p*-Value	Odds Ratio	Combined Score
Focal Adhesion WP306	3.195 × 10^−13^	1.508 × 10^−10^	1.53	43.95
Focal Adhesion-PI3K-Akt-mTOR-signaling pathway WP3932	4.242 × 10^−13^	1.001 × 10^−10^	1.42	40.59
Integrin-mediated Cell Adhesion WP185	3.947 × 10^−10^	6.210 × 10^−8^	1.63	35.32
EGF/EGFR Signaling Pathway WP437	5.413 × 10^−9^	4.258 × 10^−7^	1.47	27.94
Epithelial to mesenchymal transition in colorectal cancer WP4239	6.715 × 10^−9^	4.528 × 10^−7^	1.47	27.65
Ras Signaling WP4223	5.153 × 10^−9^	4.864 × 10^−7^	1.44	27.47
Insulin Signaling WP481	1.184 × 10^−8^	6.986 × 10^−7^	1.46	26.65
MAPK Signaling Pathway WP382	4.312 × 10^−9^	5.088 × 10^−7^	1.38	26.62
ErbB Signaling Pathway WP673	4.975 × 10^−8^	2.348 × 10^−6^	1.58	26.59
VEGFA-VEGFR2 Signaling Pathway WP3888	2.530 × 10^−8^	1.327 × 10^−6^	1.37	23.96

## Data Availability

All primary samples utilized in this study have been obtained by informed consent, in line with the Declaration of Helsinki. Data have been uploaded on NCBI GEO, accession ID GSE165011.

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
