# Peer review of "Distinct miRNA Signatures and Networks Discern Fetal from Adult Erythroid Differentiation and Primary from Immortalized Erythroid Cells"

_ijms, 2021, doi:10.3390/ijms22073626_

Round 1

Reviewer 1 Report

General comments

This manuscript presents a thorough analysis of miRNA differential expression during erythroid differentiation of human primary hematopoietic progenitor cells and, in parallel, of two human erythroid cell lines, HUDEP-1 and HUDEP-2. The generated data are important in the field of erythropoieisis as these lines are widely used in research laboratories. The results are very descriptive but they are analyzed in depth.

Major issue

There is a lack of functional data related to the results. The authors discuss a large number of differentially expressed miRNAs that are likely important in the process of erythroid differentiation and/or hemoglobin switch, but at no time they correlate differential expression with the expression levels of target genes in their cell culture model. Moreover, no study aiming to reduce or increase the expression of newly discovered miRNAs has been made to support their real implication in erythropoieisis and/or globin switch. The provision of results concerning these points would give more interest but also more credit to the study. The inclusion of figures modeling the "multiple to multiple" interaction mechanisms the authors speak about would go a long way in understanding their thinking.

Specific comments

Abstract :

“Moreover, comparison between HUDEP-2 and HUDEP-1 cells indicated miRNAs, TFs, target genes, and pathways associated with globin switching.” This sentence is not clear and may be changed as such: “comparison between HUDEP-2 and HUDEP-1 cells displayed changes in miRNAs, TFs, target genes, and pathways associated with globin switching”

  1. Introduction

Even if it is interesting, the introduction could be reduced by 30-50% and be confined to what is strictly necessary to understand the interest of the study and the results. Non-essential information (examples of differentially regulated miRNAs, description of HUDEP cells,…) may be transferred to a supplementary file.

  1. Results

- Page 3.

A diagram showing a direct comparison of cells (HUDEP Vs. CD34+) and the different stages of cell culture with the days on which samples _E and _L were taken would be highly welcome. It is necessary to explain why days 6 and 9, for primary cells and HUDEP cells respectively, were chosen for comparison. Flow cytometry and microscopy must be shown here to characterize cell stages.

- Page 4.

Which parameters were taken into account for the principal component analysis (Fig 1a)?

Were the miRNAs detected up to a few reads included in the analyzes shown figures 1b-1d? If so, what is the relevance of their existence? If not, how was the limit set?

Fig 1d: Based on this figure, it is not clear to this reviewer why HUDEP-1 and HUDEP-2 could be grouped together and separately from CD34+ cells. 

- Page 5.

Figure 2: The reason why figure 2 is included as a main figure is not clear, especially since the results of this figure are little commented on. It could be included  as a supplementary figures.

- Page 7.

“our data revealed a restricted pattern of global miRNA expression and a reciprocal increase of the expression of individual lineage-specific miRNAs in terminal stages of erythropoieisis. This finding mirrors at the miRNA level what has been observed at the mRNA levels, i.e., that HSPCs retain an uncommitted, poised cell state that is characterized by transcriptional complexity (44), and indicates active regulation of the cell state by a large repertoire of miRNAs”:

In this study, the authors do not study uncommitted HSPCs, defined as fresh CD34+ cells, but cells grown in culture and whose phenotype before erythroid differentiation, on day 0, was not characterized. This sentence is therefore incorrect.

- Page 10.

It is not clear why the authors sought to position miR genes whose expression is down or upregulated on the chromosomes (Figure 4b). What information does this provide? Figure 4b could be removed. If it is just a matter of controlling the quality of the study by correlating the number of deregulated miRNAs with the number of miRNAs per chromosome, this information can be given as an additional figures.

The authors indicate that five miRNAs of the miR-106a/363 cluster located at the Xq26.2 locus were upregulated in HUDEP2_L cells. However, only three of them are shown Fig. 4a (miR-106a-5p, miR18b-5p, and miR-363-3p). What about the other two?

Based on observation made by Harden (2017), the authors conclude that differential expression may reflect the dox-induced expression of the immortalization system. What about differential expression in HUDEP2_E and HUDEP1_E compared to CD34+_E? Do the difference support such a conclusion?

- Page 11.

The authors select a few miRNAs differentially expressed between HUDEP2_L and CD34 + _L cells to show that the differences underline the known differences between cells (E6/E7 expression, cKit down regulation and enucleation) . What about the many miRNAs they don't talk about? What genes are they targeting? What are the potential cellular pathways that are affected?

- Page 12.

“Several other miRNAs previously put forward as regulators of g-globin expression through differential expression studies in cord blood vs adult reticulocytes (e.g., miR-146a and miR-150) (28) in fetal liver vs bone marrow HSPCs (e.g., miR98-5p, miR-182-5p and miR-183-5p) (60) or through functional studies (e.g., miR23a-5p, miR-27a-5p, miR-326 and miR-34a) (62-64), were also DE between HUDEP1_L vs HUDEP2_L cells in our study”:  

Why miR-150, miR-98-5p, miR-183-5P, miR23a-5p, miR-326, miR-34a are not shown Fig 5a?

And what about the one described in the next sentences?

Could the authors provide a supplementary table with differentially expressed known miRNAs in comparison of HUDEP1_L and HUDEP2_L (similar to table S7 for novel miRNAs), and indicate among those miRNAs, those which reveal a signature associated with the Hb switching and fetal vs adult erythropoiesis?

What about the other DE miRNAs?

- Page 13.

“likely reflecting variability of expression for their shared dox-inducible HPV16-E6/E7 transgene”:

For this to be likely, it must be verified that these miRNAs are expressed at much lower level in CD34+ than in HUDEP cells (HUDEP2 and HUDEP1)

Again, it is not clear why the authors sought to position the miRNAs whose expression is down or upregulated on the chromosomes (Figure 5b).

-Page 14.

The paragraph on relations between DE miRNAs, target genes and TFs is unclear. The objective and the manner in which the interactions were established must be explained with more clarity so that the uninformed reader can understand and criticize the conclusions. It is not clear whether all the miRNAs (274 and 131) were finally included in figures 6a and 6b or not. It is not clear why the sizes of the red squares are large or small and what it means. The “connections between the three interactors” (miRNA/TF; miRNA/target genes; TF/Target genes) are not visible. The reason why TFs were excluded or dropped from the list of target genes is not understandable.

Please explain after filtering out of nodes with low degree and betweenness centrality” and “ZBTB7A and KLF10 showed the highest degree and betweenness centrality”.

“9 out of a list of 17 experimentally validated miRNAs that target BCL11A were indicated as DE in HUDEP1_L vs HUDEP2_L in our study, whereas the corresponding numbers for ZBTB7A were 35 out of 100”. Could the authors provide the list?

“In an independent analysis of experimentally validated miRNA/target-mRNA interaction networks of BCL11A and ZBTB7A, we found that miRNAs in these networks were overrerprensented…”: Please provide the experimental details.

- Page 16-20

The paragraphs on the quality of data obtained by small RNA sequencing (2.6) and on erythroid  differentiation (2.7 and 2.8) should come at the beginning of the result section.

  1. Discussion

The notion of interwoven gene regulatory network where recurrent motifs of feedback and feedforward loops exist between genes is important. Figures representing possible models of connection between TF, target genes and miRNAs, as well as the evocation of specific examples based on the study presented in this manuscript, would be very useful to help understand the discussion.

  1. Materials and methods

Cell cultures:

The exact culture conditions of the immortalized and primary cells are absolutely critical to reproducing the results. They should therefore be described with the greatest care, including when they are based on previous references.

How long were HUDEP-2 cells maintained in culture in phase III?

What is protocol C? Please describe.

How long were CD34+ cells maintained in stemspan CC100 medium before induction of erythroid differentiation? How long were cells kept in this medium?

105 M 2-Mercapto should be 10-5M.

To which stages and which days of the differentiation protocols correspond the terms CD34_E, HUDEP1_E, HUDEP2_E, CD34_L, HUDEP1_L, HUDEP2_L?

  1. Tables

Table S1: please define “known”,  “novel”, piRNA

Tables S2 and S3: Why the miRNA for which no read was detected in a sample take the value of 0.001 after normalization?

Table S7: could the authors highlight

  1. Figures

Figure 1: some elements of interpretation included in the legend also appear in the main text. They may be removed

Figure 2: The main component analysis (Fig 1a) shows that the most important parameter is the stage of culture. The heatmap in FIG. 2 would therefore better be reorganized so as to position the results obtained in early cells and then in late cells one beside the other. Ex: CD34_E then HUDEP2_E then HUDEP1_E then HUDEP1_L then HUDEP2_L then CD34_L

Figure 3, Figure 4, Figure 5: What are miRNAs in red?

Figure 6: require further explanation

Figure 8b: To the opinion of this reviewer, distinction of cell type based on morphology and determination of percentages is subject to caution based on these images. 

Figure S1: it is not clear which reads are assigned to miRNAs and how they were.

Figure S4: it is a surprise not to find the expected values for log2Ratio(_L/ _E). For example, upon erythroid differentiation, novel_mir1 goes from 2621.85 to 631.28 transcripts per million reads. Thus log2 ratio = log2 (631.28/2621.85) = -2.05, and not -1.51

Reviewer 2 Report

In their manuscript, the Authors investigated the differential expression of microRNAs in primary human peripheral-blood-derived CD34+ cells and in HUDEP-2 and HUDEP-1 immortalized cell lines, considering for each cellular model, different levels of erythroid differentiation (early and late). Authors also have examined the possible relationship between DE miRNAs, target genes and transcription factors, in order to possibly suggest TF-miRNA co-regulatory network. In particular, TF-miRNA networks and analysis of target genes of known and novel differentially expressed miRNAs, as well as the evidence of possible crosstalk between miRs and lncRNAs, are, in my opinion, very interesting. Results obtained allowed linking the differential miRNA expression (including the identification of novel miRNAs) to erythroid differentiation, cell type, and hemoglobin expression profile.

The article is nicely written all key elements have been considered. DNA Nanoball sequencing data and validation results by RT-qPCR are described completely and clearly, with a lot of figures, tables and supplementary information, and the results discussed in all paragraphs are very well presented and exhaustive. Furthermore, the Authors provided a sufficient number of references to support the paper. The quality of the English language and style is fine.

This study provides important useful insights about miRNA regulatory mechanisms in erythropoiesis and globin expression, highlighting co-regulation of miRNAs as central to their natural function. MiRNAs, genes and pathways identified through this study may be explored as potential targets for the development of novel therapies for β-hemoglobinopathies and other disorders of erythropoiesis.

Author Response

We thank this Reviewer for the time taken to evaluate our manuscript and are delighted about the positive assessment of our work.

Round 2

Reviewer 1 Report

The study is carried out in an appropriate manner and the results are analyzed thoroughly. The data generated by this study will undoubtedly be an important source of information for further research. The authors have appropriately addressed all the issues and have modified the text accordingly. The following few minor changes could be made:

1) Paragraphs 2.5, 2.6, 2.7

As part of their strategy to validate their sequencing data, the authors discuss miRNAs whose specificities in terms of expression level, expression variations between late and early erythropoiesis, expression changes between adult and fetal erythroid cells, expression specificities in transformed HUDEP cells, are known or expected. To facilitate reading and understanding of the validation strategy, I recommend to make, perhaps in the form of a table, the summary list of miRNAs used as “quality control”, and to indicate in this table, the reason why it is considered that their specificity validates the study. For better clarity, only miRNAs whose fold-change and/or high expression level are known/expected, based on the results of the literature, should be highlighted within figures 4b/4C/4D/5A/6A. A color code could be implemented depending on the reason. The other miRNAs, the presence of which is discussed for other particularities, may appear in a even different color.

2) Supplementary tables 4 and 8 are associated with figures 4 (DE miRNAs _L vs _E) and 6 (DE miRNAs HUDEP1_L vs HUDEP2_L) respectively. Supplementary tables showing the full list of data depicted figure 5 (DE miRNAs HUDEP2_L vs CD34_L) and sup. Fig. 3 (DE miRNAs HUDEP1_L vs CD34_L) may also be included.

3) “Other erythroid-specific miRNAs, such as miR-15b-5p, miR-16-5p, miR-96-5p and miR-22-3p, were also significantly upregulated in CD34_L (in line with published data 18,41,51) and HUDEP2_L cells, whereas only 15b-5p was upregulated, albeit insignificantly, in HUDEP1_L “: Whereas all four miRNAs are highlighted in red Fig4B, miR-22-3p is not shown Fig 4D.  

4) “in fetal liver vs bone marrow HSPCs (e.g., miR-98-5p, miR-182-5p and miR-183-5P)”: I could not find miR-183-5p in figure 6.

5) Fig. 4a: The terms upregulated and downregulated are not easy to interpret (compared to what?). It may be wise to recall in which condition expression is up or downregulated (X or Y in “X vs Y”).

6) Please include “response 22” (made during the previous review) in the legend of figure 7:  “Analyses were performed in line with miRNet recommendations to limit network size below 2000 nodes. To do so, the “Data Filter” function was used, based on the network topology measures degree, betweenness, and shortest path. In this context, the degree of a node is the total number of connections it has to other nodes, and high-degree nodes are considered important “hubs” in a network (set to the default 1.0 here). The betweenness measures the number of shortest paths going through a node, and nodes with higher betweenness are important interactors in a network (set to 1.0 here). The shortest path filter helps reduce complexity of a network by keeping only one, shortest path between hub nodes, i.e. in the presence of multiple paths connecting two nodes, only the shortest path will be retained (applied here).”

7) Sup. Figure 1: could the authors write “miRNA” after “mature”

8) Sup. Figure 5: title is missing in the supplementary material section. Figure numbering is wrong in the figure legend.
